# Acute Chagas disease in Brazil from 2001 to 2018: A nationwide spatiotemporal analysis

**Emily F. Santos[1⊙], Ângelo A. O. Silva[1⊙], Leonardo M. Leony[1], Natália E. M. Freitas[1], Ramona T. Daltro[1], Carlos G. Regis-Silva[1], Rodrigo P. Del-Rei[2], Wayner V. Souza[3], Alejandro L. Ostermayer[4], Veruska M. Costa[5], Rafaella A. Silva[5], Alberto N. Ramos Jr[6], Andrea S. Sousa[7,8,9], Yara M. Gomes[3,9], Fred L. N. Santos[1,9]***

**1** Gonçalo Moniz Institute, Oswaldo Cruz Foundation, Salvador, Bahia, Brazil, **2** Faculty of Technology and Sciences of Bahia, Salvador, Bahia, Brazil, **3** Aggeu Magalhães Institute, Oswaldo Cruz Foundation, Recife, Pernambuco, Brazil, **4** Chagas Disease Study Center, Federal University of Goiás, Goiânia, Goiás, Brazil, **5** General Coordination of Zoonosis and Vector Diseases, Secretariat of Health Surveillance (SVS), Brazilian Ministry of Health, Brasília, Federal District, Brazil, **6** Department of Community Health, School of Medicine, Federal University of Ceará, Fortaleza, Ceará, Brazil, **7** Federal University of Rio de Janeiro, School of Medicine, Department of Internal Medicine, Rio de Janeiro, Rio de Janeiro, Brazil, **8** Laboratory of Clinical Research on Chagas Disease, Evandro Chagas National Institute of Infectious Disease, Oswaldo Cruz Foundation, Rio de Janeiro, Brazil, **9** Chagas Disease Translational Research Program (Fio-Chagas), Oswaldo Cruz Foundation, Rio de Janeiro, Rio de Janeiro, Brazil

⊙ These authors contributed equally to this work.
* fred.santos@fiocruz.br

**Data Availability Statement:** The data underlying the results presented in the study are available from the Data Information Department of the Unified Health System (DATASUS) (http://www2.

## Abstract

### Background

In Brazil, acute Chagas disease (ACD) surveillance involves mandatory notification, which allows for population-based epidemiological studies. We conducted a nationwide population-based ecological analysis of the spatiotemporal patterns of ACD notifications in Brazil using secondary surveillance data obtained from the Notifiable Diseases Information System (SINAN) maintained by Brazilian Ministry of Health.

### Methodology/Principal findings

In this nationwide population-based ecological all cases of ACD reported in Brazil between 2001 and 2018 were included. Epidemiological characteristics and time trends were analyzed through joinpoint regression models and spatial distribution using microregions as the unit of analysis. A total of 5,184 cases of ACD were recorded during the period under study. The annual incidence rate in Brazil was 0.16 per 100,000 inhabitants/year. Three statistically significant changes in time trends were identified: a rapid increase prior to 2005 (Period 1), a stable drop from 2005 to 2009 (Period 2), followed by another increasing trend after 2009 (Period 3). Higher frequencies were noted in males and females in the North (all three periods) and in females in Northeast (Periods 1 and 2) macroregions, as well as in individuals aged between 20–64 years in the Northeast, and children, adolescents and the elderly in the North macroregion. Vectorial transmission was the main route reported during Period 1, while oral transmission was found to increase significantly in the North during the other periods. Spatiotemporal distribution was heterogeneous in Brazil over time. Despite regional

datasus.gov.br/DATASUS) and from the Brazilian Institute of Geography and Statistics (IBGE), based on the national census for the period between 2000 and 2010 (https://sidra.ibge.gov.br/pesquisa/censo-demografico/series-temporais/series-temporais/).

**Funding:** This work was supported by the Gonçalo Moniz Institute, Coordination of Superior Level Staff Improvement-Brazil (CAPES) - Finance Code 001 and Research Support Foundation of the State of Bahia (FAPESB). Wayner Vieira de Souza is research fellows of CNPq (process no. 306222/2013-2). The funders had no role in study design, data collection and analysis, decision to publish, or preparation of the manuscript.

**Competing interests:** The authors have declared that no competing interests exist.

differences, over time cases of ACD decreased significantly nationwide. An increasing trend was noted in the North (especially after 2007), and significant decreases occurred after 2008 among all microregions other than those in the North, especially those in the Northeast and Central-West macroregions.

## Conclusions/Significance

In light of the newly identified epidemiological profile of CD transmission in Brazil, we emphasize the need for strategically integrated entomological and health surveillance actions.

## Author summary

Chagas disease (CD) infection is a debilitating and neglected disease that occurs in 21 Latin America countries. CD has two distinct phases: acute and chronic. The generally asymptomatic acute phase begins shortly after infection and can last up to four months. When symptoms do appear, they are typically mild and unspecific. Following this phase, infected individuals evolve to a long-lasting chronic phase, which can be either symptomatic or asymptomatic. In Brazil, only acute cases are mandatorily notifiable in the Brazilian Notifiable Diseases Information System (Brazilian Ministry of Health). Most chronic cases are unknown and untreated. Considering that epidemiological data related to ACD is publicly available, we have analyzed the spatiotemporal distribution of notified cases of ACD and evaluated relevant epidemiological indicators throughout Brazil from 2001 to 2018. The data present here may contribute to surveillance actions designed at preventing new CD cases. We observed 5,184 cases of ACD during the period under study. The annual incidence rate in Brazil was 0.16 per 100,000 inhabitants/year. Three distinct epidemiological periods were identified: a rapid increase prior to 2005 (Period 1), a stable drop from 2005 to 2009 (Period 2), followed by another increasing trend after 2009 (Period 3). Vectorial transmission was the main route reported during Period 1, while oral transmission was found to increase significantly in the North during the other periods. Despite regional differences, over time cases of ACD decreased significantly nationwide. An increasing trend was noted in the North (especially after 2007). In light of the newly identified epidemiological profile of CD transmission in Brazil, we emphasize the need for strategically integrated entomological and health surveillance actions.

## Introduction

Chagas disease (CD) is an anthropozoonosis caused by the hemoflagellated kinetoplastid *Trypanosoma cruzi*. The disease is endemic in 21 Latin American countries, affecting approximately 6–8 million people and generating an average of 14,000 deaths annually [1,2]. The epidemiological pattern of CD has undergone substantial changes in recent decades as *T. cruzi*-infected migrants from endemic areas have moved into non-endemic regions in the North America, Europe, Asia and Oceania [3–5].

CD has two distinct phases. The generally asymptomatic acute phase begins shortly after infection and can last up to 4 months. When symptoms do appear, they are typically mild and unspecific [6]. Although sometimes higher in children, the risk of death during acute phase

can reach 5% and is generally related to complications associated with both meningoencephalitis and/or myocarditis [7,8]. Following this phase, infected individuals evolve to a long-lasting chronic phase, which can be either symptomatic or asymptomatic [9].

In Brazil, CD persists as a relevant public health problem [10]. This disease was the leading cause of disability-adjusted life years (DALYs) among all Neglected Tropical Diseases (NTD), followed by schistosomiasis and dengue [11]. Due to a past history of vectorial transmission that was later virtually interrupted [12], more individuals in Brazil are in the chronic phase of CD. Despite the prevalence of the chronic form, only acute cases are mandatorily notifiable in the Brazilian Notifiable Diseases Information System (*Sistema de Informação de Agravos de Notificação* SINAN, Brazilian Ministry of Health). Between 2003 and 2018, 4,556 cases of ACD were reported, with changes in the affected macroregions occurring after 2007, as higher concentrations of cases and incidence were reported in the North [13]. A nationwide study based on an analysis of the Mortality Information System between 1999 and 2007 revealed that CD was noted on 53,930 (0.6%) death certificates, and that ACD was listed as the cause of death in 2.8% [14].

Considering that epidemiological data related to CD is publicly available, analysis can aid in the prioritization of regional epidemiological disease surveillance efforts. With the aim of contributing to surveillance actions designed at preventing new CD cases, our results detail the spatiotemporal distribution of notified cases of ACD and evaluate relevant epidemiological indicators throughout Brazil from 2001 to 2018.

## Materials and methods

### Study area

The present study was conducted in Brazil, the largest country in Latin America, with the world's fifth largest geographic area and population: over 207 million residents at a density of 41 inhabitants/km$^2$ (2017). Brazil's territory also extends into much of the continent's interior and borders other countries reporting a high prevalence of CD. Politically and administratively, Brazil is divided into 26 states and one Federal District. The Federation is further grouped into five macroregions (North, Northeast, Southeast, South and Central-West) and 558 microregions containing 5,567 municipalities with differing geographic, socioeconomic and cultural characteristics (Fig 1).

### Study population and design

This nationwide population-based ecological study was based on secondary surveillance data, and employed a spatiotemporal analysis of ACD notifications aggregated according to microregion. Chagas disease surveillance encompasses the compulsory notification to SINAN of all confirmed cases of ACD; all cases reported between 2001 and 2018 in the 558 Brazilian microregions were included. The SINAN database is publicly accessible and data is available online by the Data Information Department of the Unified Health System (DATASUS) (http://www2.datasus.gov.br/DATASUS). SINAN aggregates information on indicators related to priority diseases in Brazil, as is used to support control actions. To investigate temporal trends in CD, we obtained records for all ACD notifications from the SINAN database organized according to each Brazilian microregion. Other variables, such as age, gender, ethnicity and probable route of infection were also analyzed using this data. To estimate infection rates, population data were obtained from the Brazilian Institute of Geography and Statistics (IBGE), based on the national census for the period between 2000 and 2010 (https://sidra.ibge.gov.br/pesquisa/censo-demografico/series-temporais/series-temporais/), while official annual

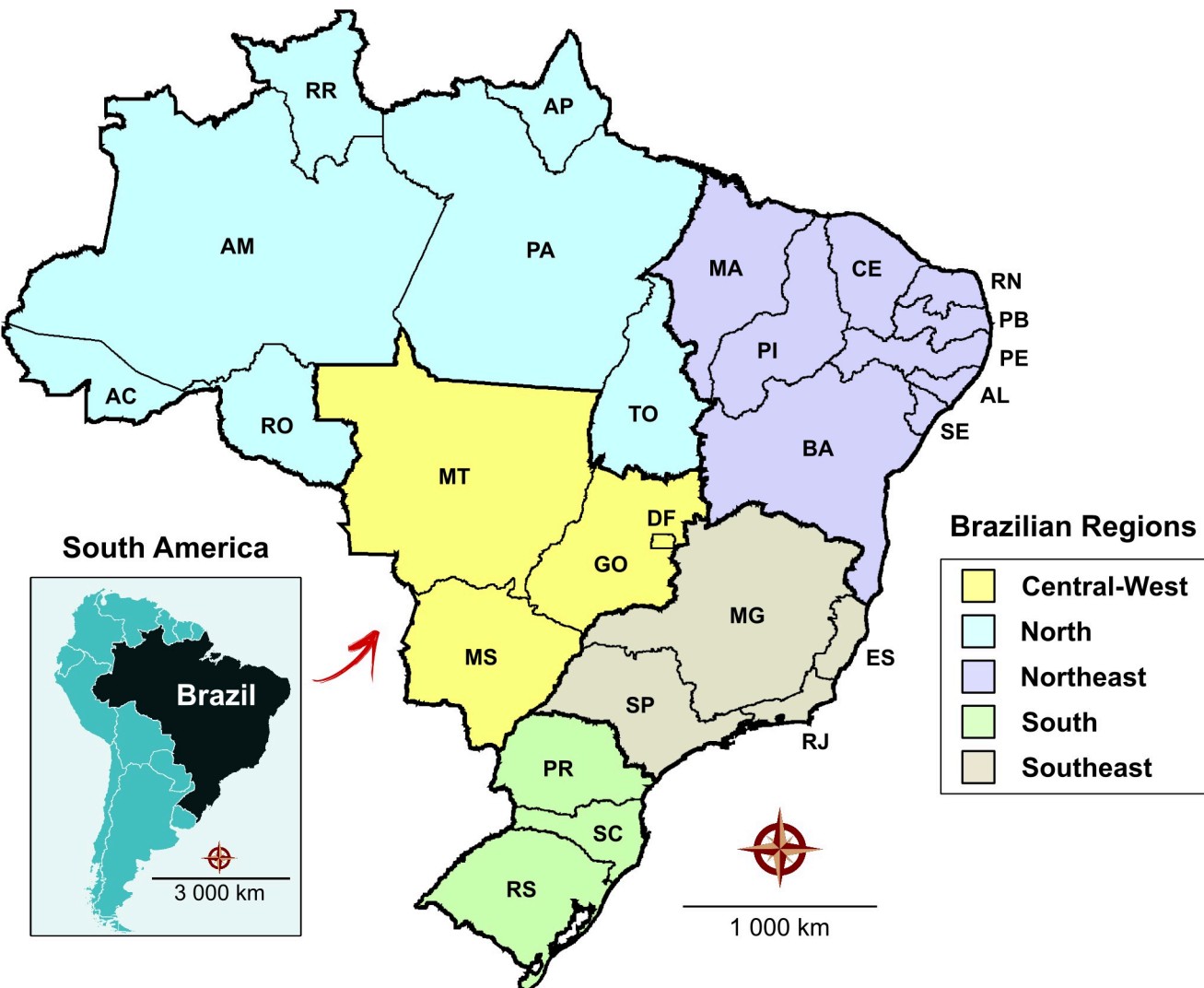

**Fig 1. Brazil is geographically divided into five macroregions, and administratively into 26 states and one Federal District (DF).** Central-West (DF: Distrito Federal, GO: Goiás, MT: Mato Grosso and MS: Mato Grosso do Sul); North (AC: Acre, AM: Amazonas, AP: Amapá, RO: Rondônia and RR: Roraima); Northeast (AL: Alagoas, BA: Bahia, CE: Ceará, MA: Maranhão, PB: Paraíba, PE: Pernambuco, PI: Piauí, RN: Rio Grande do Norte and SE: Sergipe); South (PR: Paraná, RS: Rio Grande do Sul and SC: Santa Catarina); Southeast (ES: Espírito Santo, MG: Minas Gerais, RJ:Rio de Janeiro and SP: São Paulo). Public domain digital maps were obtained from the Brazilian Institute of Geography and Statistics (IBGE) cartographic database in shapefile format (.shp), which was subsequently reformatted and analyzed using QGIS version 3.10 (Geographic Information System, Open Source Geospatial Foundation Project. http://qgis.osgeo.org).

population estimates were used for the remaining years (available at https://sidra.ibge.gov.br/pesquisa/estimapop/tabelas).

## Data analysis

Spatial analyses were performed to identify the spatial distribution of the variables related to CD notifications. All ACD notifications reported on a municipality level, specifically linked to the municipality of residence of each CD case, were grouped into microregions, which were then used as a unit of analysis to compare among different regions in order to reveal priority areas for interventions. Three-year moving averages were calculated between 2001 and 2018 [15]. Annual age- and sex-adjusted incidence rates with corresponding 95% confidence

intervals (CI) were calculated per 100,000 inhabitants using population census data from 2010 and annual population estimates. Temporal trends in adjusted annual incidence rates were calculated employing joinpoint regression models [16], stratified according to microregion. For this analysis, each joinpoint indicated a statistically significant change in the slope tested using Monte Carlo permutation testing. Annual percentage changes (APC) and 95% CIs were calculated for each segment. Trends were considered statistically significant when APC presented a p-value < 0.05. Maps were created using the Brazilian annual incidence at the beginning of the studied period as a denominator to illustrate the relative risk of ACD among the country's microregions. Mapping was done with QGIS software version 3.10 (Geographic Information System, Open Source Geospatial Foundation Project; freely available at: http://qgis.osgeo.org). Digital maps were obtained from the IBGE database in shape file (.shp) format, compatible with the QGIS program. A checklist (S1 Checklist) is provided according to the Strengthening the Reporting of Observational studies in Epidemiology (STROBE) guidelines [17].

## Ethics

SINAN and IBGE databases, which are available in the public domain, do not allow for the identification of individuals. In 2016, a new resolution published by the Brazilian National Health Council abrogated the need to seek approval from any Institutional Review Board for studies using publicly available secondary data that does not provide individually identifiable information (http://conselho.saude.gov.br/resolucoes/2016/reso510.pdf).

## Results

### Spatial distribution of acute CD

Between 2001 and 2018, 5,184 cases of ACD were reported to SINAN. At least one case of ACD was reported in 307 of 558 (55.0%) microregions, corresponding to 3,238 municipalities and a total population size of 158,363,480 inhabitants (76.3% of the Brazilian population). The annual rate of reported cases in Brazil between 2001 and 2018 was 0.16 per 100,000 inhabitants/year, ranging from 0.07 to 0.32 notified cases per 100,000 inhabitants. Joinpoint regression analysis from 2001 to 2018 revealed three statistically significant changes in tendency: a rapid increase in cases from 2001 to 2005, a considerable drop from 2005 to 2009, followed by another increase during the period from 2009 to 2018 (Fig 2). Based on these findings, the evaluated acute CD notifications were divided into three distinct periods: Period 1 (corresponding to APC 1), Period 2 (APC 2) and Period 3 (APC 3).

Sociodemographic variables, self-reported skin color and probable route of *T. cruzi* infection of all ACD notifications are summarized in Table 1. Of the 5,184 cases of ACD, 2,607 (50.29%) were male and 2,575 (49.67%) female. Most cases (70.68%) were middle-aged adults from 20 to 64 years. However, infected cases were also found in all ages. In the total sample, self-reported skin color was presented in the following proportions of cases: brown/mixed-race (60.59%), white (20.71%), black (7.25%), indigenous (0.98) and yellow (0.79%). According to the probable route of infection, overall, vectorial (35.39%) and oral (38.27%) were the most important. It is noteworthy that the North region is responsible for 55.62% of cases in Brazil.

Fig 3 illustrates changes in the profiles of the sociodemographic variables studied during the three periods identified (data available in S1 Table). In the North macroregion, ACD was similarly reported in males and females. Conversely, in the Northeast, infection was more frequently found in women during Periods 1 and 2. With respect to age, most reported cases were concentrated in individuals aged between 20 and 64 years. However, during Periods 2 and 3, in contrast to the rest of the country, reports of ACD in the North macroregion were more frequent in children (≤9 years), adolescents (10–19 years) and elderly individuals (≥65

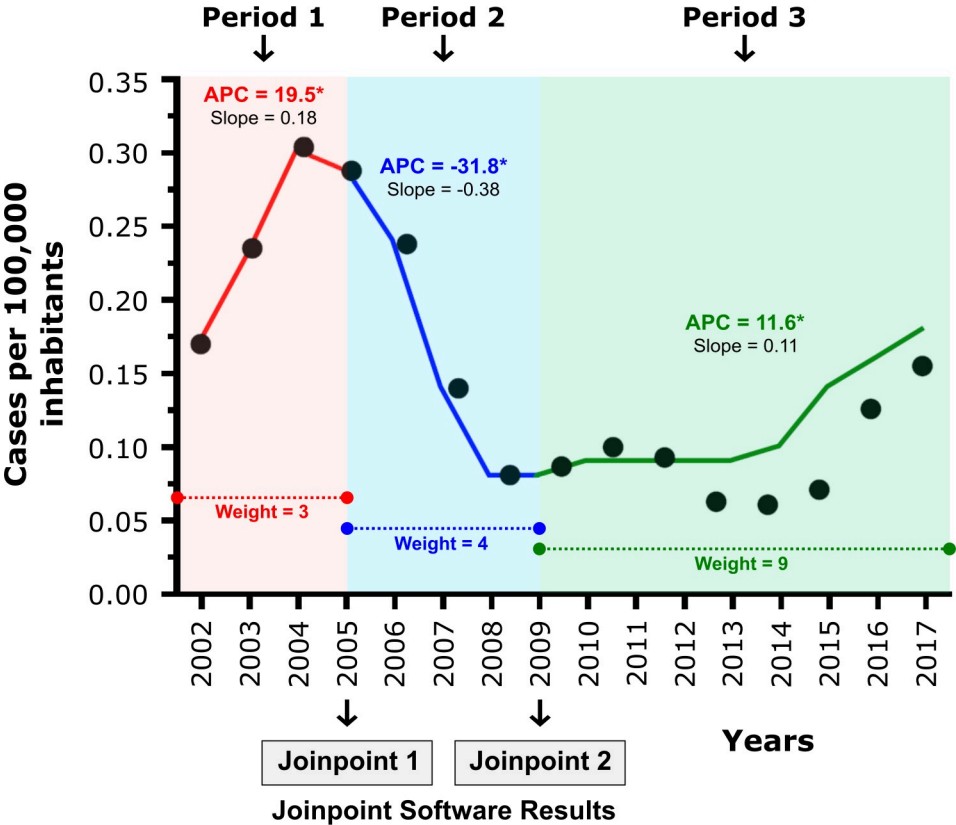

**Fig 2. Global rates (expressed at 3-year moving averages) of ACD notifications in Brazil per 100,000 inhabitants (2002 to 2017) according to SINAN.** Three periods were identified by Joinpoint regression using APC (Annual Percentage Change) calculations: Period 1 (2001–2005), Period 2 (2005–2009) and Period 3 (2009–2018).

years). The miscegenation of the Brazilian population is reflected by the fact that most reported cases were attributed to individuals who self-identified ethnicity as 'brown', except during Periods 1 and 2 in the South, where CD was more prevalent in those who self-identified skin color as white. The variations observed in the probable mode of infection evidence a change in the epidemiological profile of CD transmission: Vectorial transmission was the main route reported during Period 1, while oral transmission increased significantly in the North as a result of many case notifications originating from the states of Pará (Furos de Breves—80 cases; Belém—232 cases; Cametá - 88 cases) and Amapá (Macapá - 93 cases) in Period 2, as well as in Pará (Portel—63 cases; Furo de Breves—359 cases; Belém—848 cases; Cametá - 500 cases) in Period 3. Unfortunately, mainly in the Northeast and North macroregions, data on the probable route of transmission and self-reported skin color were frequently missing from notification records in the SINAN database.

The spatiotemporal distribution of reported rates of ACD infection is shown at 16 distinct time points in Fig 4 (data available in S2 Table). Overall, significant changes in the incidence of CD throughout Brazil are evidenced over time. The number of positive cases was found to increase in the state of Pará (North macroregion), especially after 2007. Indeed, in 2016, high numbers of cases were reported in the microregions of Furos de Breves (170.78 cases per 100,000 inhabitants), Cametá (80.86 cases per 100,000 inhabitants), Portel (63.14 cases per 100,000 inhabitants) and Belém (28.16 cases per 100,000 inhabitants). Similarly, increasing numbers of cases were also notified in the state of Acre (North region) after 2014, as evidenced

**Table 1. Data stratified by sociodemographic variables, self-reported skin color and probable route of *Trypanosoma cruzi* infection during the period under study (2001–2018) of acute Chagas disease notifications\*.**

| Category | Division | Frequency | Percentage (%) |
|---|---|---|---|
| Gender classification | Male | 2,607 | 50.29 |
| | Female | 2,575 | 49.67 |
| | No data | 2 | 0.04 |
| Age classification | ≤ 4 | 213 | 4.11 |
| | 5–9 | 256 | 4.94 |
| | 10–19 | 572 | 11.03 |
| | 20–64 | 3,664 | 70.68 |
| | ≥ 65 | 475 | 9.16 |
| | No data | 4 | 0.08 |
| Self-reported skin color | White | 1,073 | 20.71 |
| | Black | 376 | 7.25 |
| | Yellow | 41 | 0.79 |
| | Brown | 3,141 | 60.59 |
| | Indigenous | 51 | 0.98 |
| | No data | 502 | 9.68 |
| Probable route of infection | Vectorial | 1,834 | 35.39 |
| | Mother-to-child | 19 | 0.37 |
| | Accidental | 4 | 0.08 |
| | Oral | 1,984 | 38.27 |
| | Transfusional | 16 | 0.31 |
| | No data | 1,327 | 25.60 |
| Year of notification | 2001 | 59 | 1.14 |
| | 2002 | 246 | 4.75 |
| | 2003 | 579 | 11.17 |
| | 2004 | 419 | 8.08 |
| | 2005 | 629 | 12.13 |
| | 2006 | 544 | 10.48 |
| | 2007 | 156 | 3.01 |
| | 2008 | 104 | 2.01 |
| | 2009 | 220 | 4.24 |
| | 2010 | 130 | 2.51 |
| | 2011 | 190 | 3.67 |
| | 2012 | 189 | 3.65 |
| | 2013 | 163 | 3.14 |
| | 2014 | 196 | 3.78 |
| | 2015 | 268 | 5.17 |
| | 2016 | 372 | 7.18 |
| | 2017 | 340 | 6.56 |
| | 2018 | 380 | 7.33 |
| Brazilian regions | Central-West | 112 | 2.16 |
| | North | 2,883 | 55.62 |
| | Northeast | 1,673 | 32.27 |
| | South | 291 | 5.61 |
| | Southeast | 225 | 4.34 |

\*Data updated on May 22, 2020.

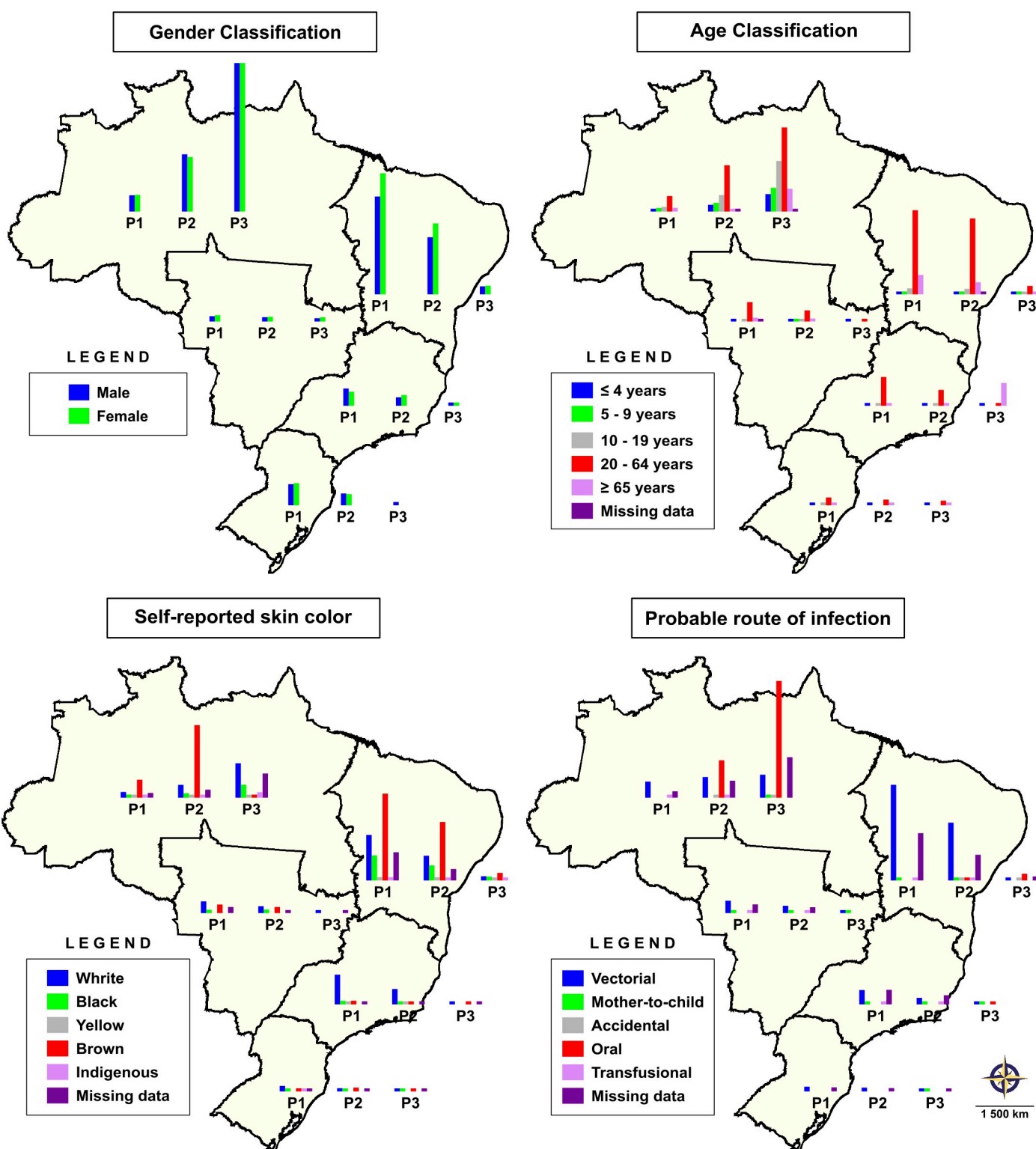

**Fig 3. Analysis of changes in profiles of sociodemographic variables, self-reported skin color and probable route of *Trypanosoma cruzi* infection by region, stratified according three periods (P1, P2 and P3) of acute Chagas disease notifications as determined by annual percentage changes.** Public domain digital maps were obtained from the Brazilian Institute of Geography and Statistics (IBGE) cartographic database in shapefile format (.shp), which was subsequently reformatted and analyzed using QGIS version 3.10 (Geographic Information System, Open Source Geospatial Foundation Project. http://qgis.osgeo.org).

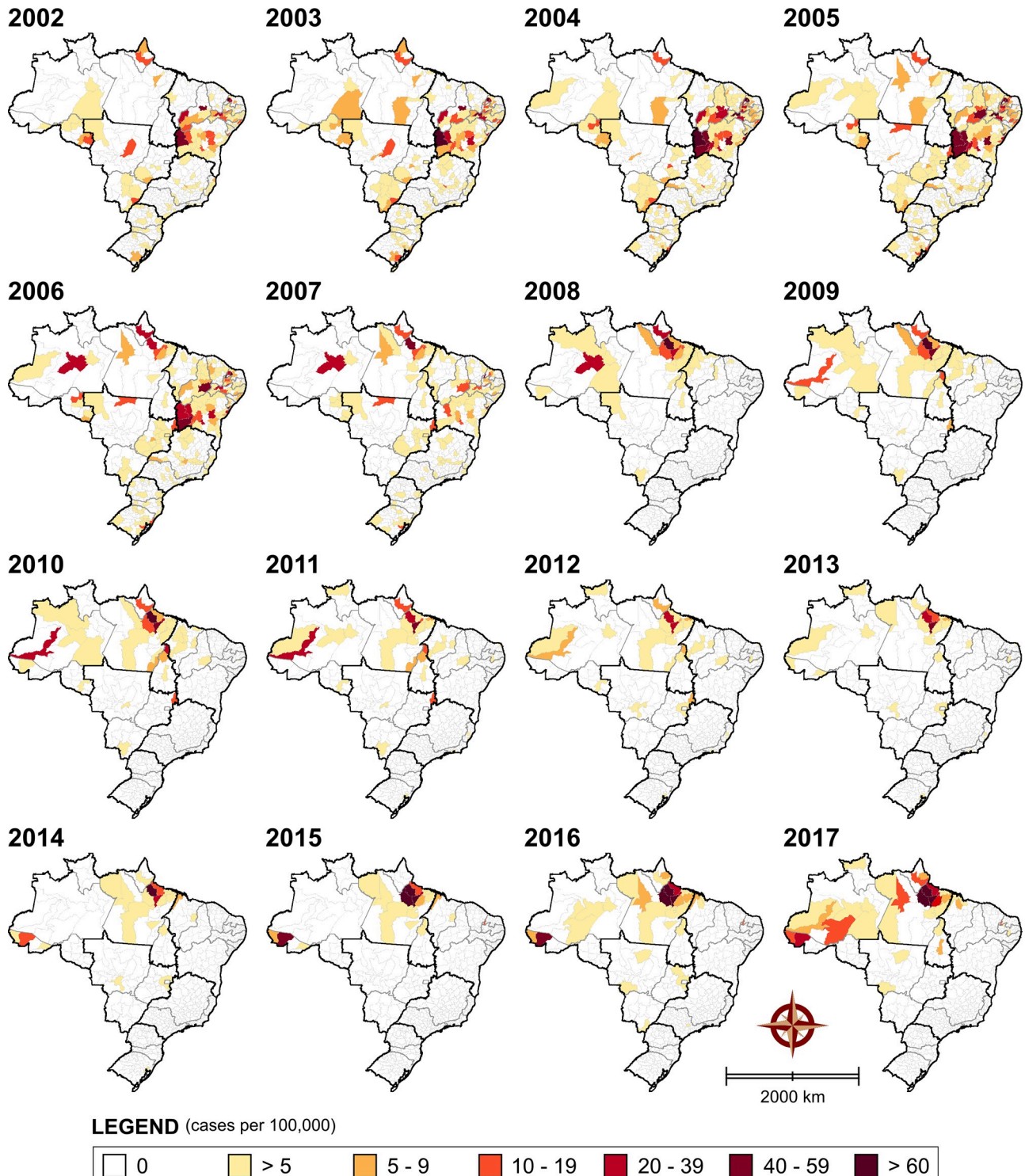

**Fig 4. Spatiotemporal distribution of relative risk of ACD by microregion, based on number of case notifications (SINAN-Brazilian Ministry of Health, Brazil, 2002–2017).** Public domain digital maps were obtained from the Brazilian Institute of Geography and Statistics (IBGE) cartographic database in shapefile format (.shp), which was subsequently reformatted and analyzed using QGIS version 3.10 (Geographic Information System, Open Source Geospatial Foundation Project. http://qgis.osgeo.org).

in the Tarauacá microregion in 2016 (44.75 cases per 100,000 inhabitants). Conversely, beginning in 2008, significant decreases were seen in the number of cases among all microregions other than those in the North, especially in the Northeast and Central-West macroregions. Interestingly, after 2007, a dramatic drop in the number of acute case notifications occurred, and, in some areas previously considered highly endemic for CD, no cases were reported. In Bahia, this was particularly true in some microregions that presented high rates of cases per 100,000 inhabitants before 2007, such as those in the western part of the state: Barreiras (102.64 in 2004), Santa Maria da Vitória (48.08 in 2005) and Cotegipe (42.8 in 2005). Similarly, Itaberaba (Bahia), a microregion located in Chapada Diamantina, a semi-arid area in the state's central region, presented 45.64 cases per 100,000 inhabitants in 2004. Other microregions with a similar climate located in the Brazilian Northeast, e.g. Pau dos Ferros and Médio Oeste (Rio Grande do Norte) reported 73.51 and 112.36 cases per 100,000 inhabitants in 2005, respectively; Alto Médio Canindé (Piauí) notified 42.60 cases per 100,000 inhabitants in 2005, and Salgueiro (Pernambuco), 65.32 cases per 100,000 inhabitants in 2004.

## Discussion

The present findings indicate that acute Chagas disease continues to present a threat to public health in Brazil, as evidenced by the occurrence of acute cases in more than 50% of Brazilian microregions. The persistence of acute cases was particularly notable in the North macroregion of Brazil in recent years, especially those increasingly associated with oral transmission. Throughout the period studied, three distinct epidemiological moments were identified, revealing both increasing and decreasing trends in numbers of reported cases. Between 1990 and 2006, the Southern Cone Initiatives in Latin America led to significant changes in the epidemiological landscape of CD [10]. In Brazil, for example, advances in the control of the main domiciliary vector (*Triatoma infestans*) were paramount to the epidemiological shift witnessed in recent decades [18]. As a direct consequence, the reduction in vector incidence contributed to a significant decline in acute CD case notifications, as seen in Periods 2 and 3. However, despite the efforts of successful prevention campaigns conducted in the country since 2000, it is possible that the decreasing incidence observed could also be related to the underreporting of cases. The decrease in notifications observed in Period 3 was concomitant with the progressive adaptation of triatomine insects to the vacant domestic and peridomestic niches that emerged as a result of *T. infestans* elimination programs, particularly in areas where sylvatic and peridomestic species occur concomitantly. The rise of new triatomine species, such as *Panstrongylus megistus*, *Triatoma brasiliensis*, *Triatoma pseudomaculata* and *Triatoma sordida*, among others, highlights the complex situation faced by Brazil, which thusly necessitates permanent surveillance [12].

The intrinsic epidemiological characteristics of NTDs can be probed through the analysis of distribution throughout specific areas. Despite the statistical similarities observed between genders [19], our findings revealed that both women and men were infected in the North of Brazil, whereas more women were infected in the Northeast. A previous study identified a predominance of acute CD in men (72.09%) and attributed this to increased contact with natural vector habitat, consequently traceable to occupational exposure [20]. Similarly, in the state of Maranhão, Cutrim et al. [21] found that males were more infected and suggested a sylvatic transmission cycle, as men are more likely to enter forested areas to hunt or cultivate crops, thusly increasing exposure to the vector. Furthermore, the precarious living conditions witnessed in the Brazilian Northeast also influence disease dissemination. In rural areas, for example, houses made from clay, wood and straw can shelter triatomines in the cracks and fissures of wall, thereby exposing inhabitants to CD vectors. Accordingly, studies have revealed

that household conditions also present a risk of infection, which may explain the higher incidence seen in women in northeastern Brazilian due to increased exposure in peridomestic settings [22].

The present study found a high number of cases among individuals of productive age (15–59 years), suggesting increased frequency of exposure in this population due to occupational activities that increase the risk of transmission [23]. In the Brazilian Northeast macroregion, this age group was associated with a high number of cases arising from vectorial transmission, possibly due to frequent triatomine exposure of this rural population during work-related activities [24–26]. This finding is relevant as morbidity and mortality in acute CD is directly associated with reduced quality of life in infected individuals. Considering its impact on workforce productivity associated with premature disability and death, CD alone is estimated to subtract 426,000 years [27] from DALYs in the Americas, which highlights why it is considered one of the most important neglected parasitic diseases throughout the Americas. Furthermore, epidemiological and transmission models of CD predict an annual global impact of $24.73 billion dollars in health-care costs and 29,385,250 DALYs [28].

In recent decades, oral transmission has gained epidemiological relevance and is now considered one of the most important routes of infection in Brazil [29,30]. Records show that increasing number of cases have been associated with this route throughout the country since 2005, although oral transmission have been reported in the Amazon region since the 1960s. Historically, larger number of cases associated with oral transmission have been reported in the northern region of Brazil [31,32]. In fact, most acute CD outbreaks in the Amazon region have been attributable to oral transmission [30,33–36], while vectorial transmission, the second leading route of infection, is predominantly linked to occupational exposure. It is noteworthy that the state of Pará is responsible for 81% of cases arising from oral transmission in the North region, with higher proportions of cases occurring between August and February, after *açaí* (*Euterpe oleracea*) and *bacaba* (*Oenocarpus bacaba*) harvest season [31]. The consumption of contaminated food by metacyclic forms of *T. cruzi* has encouraged the adoption of sanitary practices, such as pasteurizing *açaí* juice (82.5˚C for 1 min) and blanching fruits (70± 1˚C for 10 s) from endemic areas, which was shown to efficiently eliminate *T. cruzi* in food matrices [37]. Furthermore, controlling the transport of untreated juice and other products to surrounding regions in Brazil, as well as export to other countries, may contribute to reductions in this form of newly considered foodborne disease (17).

In spite of the vector control and management measures implemented by the CD Control Program (PCDCh) established in Brazil in 1975, the Southern Cone Initiative, launched in 1991 and granted certification in 2006, is recognized as being responsible for the interruption of CD through the near-elimination of its most important domestic vector, *T. infestans* [12,38]. Despite this, some studies have shown the persistence of the vectorial route of transmission, most likely due to the existence of wild vectors in extradomicilar settings. This highlights the fact that vectorial transmission continues to present risk, possibly due to the existence of autochthonous vectors with high colonization potential, as well as residual isolated foci of *T. infestans* [38,39]. From 2012 to 2016, the Ministry of Health reported *T. infestans* colonization in four municipalities in the state of Bahia and 12 in Rio Grande do Sul [40]. To this end, several studies have described the persistence of triatomine species in cities throughout Bahia [41,42]. In sum, the persistence of vectorial transmission reinforces the need for enhanced entomological surveillance involving routine domicile visitations by public health authorities, as well as targeted educational campaigns in affected communities, chemical-based control measures, and improvements in housing conditions in areas where *T. cruzi* is endemic to effective reduce Chagas disease infection risk.

The high incidence of vectors reported in the Northeast, Southeast and South macroregions may be related to the presence of *Panstrongylus megistus*, since this vector is currently considered one of the most important in Brazil, with widespread distribution from the South to Northeast. Great microgeographic diversity has been observed in the states of Minas Gerais and Bahia, and *P. megistus* is often found in the southern, southeastern and northeastern riverside areas of the country. In addition, this data may also reflect possible underreporting, especially with respect to vectorial transmission, as many cases are not correctly identified with this route [30].

This study is mainly limited by the quality of the data analyzed, as well as the underreporting of acute CD cases and isolated outbreaks. Indeed, some of the differences observed among microregions may be the result of underreporting and/or logistical issues in local health systems, including a lack of access to specialized services in many municipalities, which provokes patients to seek care in surrounding urban centers [14]. This phenomenon likely prevents regional health professionals from properly diagnosing Chagas disease and can lead to the implementation of ineffective epidemiological surveillance. Moreover, acute cases oftentimes go undiagnosed due to a lack of classical symptoms, or when mild and unspecific [6], which can contribute to underreporting. Data regarding the probable mode of *T. cruzi*-infection and some sociodemographic conditions, such as self-reported skin color, age and gender, were missing from a substantial number of records. Our data further suffers limitations due to a change in the record format used by the SINAN database after 2006. Prior to this year, oral transmission was not an option in the field corresponding to probable mode of transmission, which was included beginning in 2007. Outbreaks have occurred in the country during the period of study, which were responsible for isolated ACD cases in Brazil [7,33,36,43–46]. The main impact of outbreaks, especially in cases of oral transmission in the North Brazilian macroregion, refers to distortion of geographical analysis Despite these drawbacks, we nonetheless consider the results presented herein to be of high validity and highly representative, since all cases of acute CD reported during the period from 2001 to 2018 were included in Brazil, a country of substantial size.

In conclusion, this study shows that most cases of acute CD in Brazil occurred mainly due to oral and vector transmission. Moreover, a more recent epidemiological profile of transmission demonstrates that, despite improvement as a result of the control of *T. infestans*, new measures should be incorporated into entomological surveillance programs to address the presence of different autochthonous vector species. In addition, hygienic-sanitary measures should also be adopted in an effort to further reduce oral transmission. We also emphasize the importance of notifying suspected cases in primary care and health service settings, as well as the performance of early diagnosis and common intermittent gaps in treatment availability as relevant factors in disease control. Similarly, notifying chronic cases in Brazil as well as identifying positive chronic cases from blood banks should be strongly considered [47–49]. The prevention, control and identification of the main risk factors associated with acute Chagas disease can lead to efficacious actions focused on differing epidemiological contexts in each affected region.

## Supporting information

**S1 Checklist. STROBE Checklist.**
(DOCX)

**S1 Table. Acute Chagas disease notifications according to sociodemographic variables, self-reported skin color and probable route of *Trypanosoma cruzi* infection by Brazilian**

**region, stratified according three periods (P1, P2 and P3).**
(XLSX)

**S2 Table. Relative risk of acute Chagas disease by microregion.**
(XLSX)

## Acknowledgments

We acknowledge Andris K. Walter for the English language revision and manuscript copyediting assistance.

## Author Contributions

**Conceptualization:** Yara M. Gomes, Fred L. N. Santos.

**Data curation:** Fred L. N. Santos.

**Formal analysis:** Emily F. Santos, Ângelo A. O. Silva, Leonardo M. Leony, Natália E. M. Freitas, Rodrigo P. Del-Rei, Wayner V. Souza, Alejandro L. Ostermayer, Veruska M. Costa, Alberto N. Ramos Jr, Andrea S. Souza, Fred L. N. Santos.

**Funding acquisition:** Fred L. N. Santos.

**Investigation:** Emily F. Santos, Ramona T. Daltro, Yara M. Gomes, Fred L. N. Santos.

**Methodology:** Emily F. Santos, Leonardo M. Leony, Natália E. M. Freitas, Ramona T. Daltro, Fred L. N. Santos.

**Project administration:** Fred L. N. Santos.

**Resources:** Fred L. N. Santos.

**Supervision:** Fred L. N. Santos.

**Validation:** Emily F. Santos, Fred L. N. Santos.

**Visualization:** Emily F. Santos, Ângelo A. O. Silva, Fred L. N. Santos.

**Writing – original draft:** Emily F. Santos, Ângelo A. O. Silva, Leonardo M. Leony, Natália E. M. Freitas, Ramona T. Daltro, Carlos G. Regis-Silva, Rodrigo P. Del-Rei, Wayner V. Souza, Alejandro L. Ostermayer, Veruska M. Costa, Rafaella A. Silva, Alberto N. Ramos Jr, Yara M. Gomes, Fred L. N. Santos.

**Writing – review & editing:** Wayner V. Souza, Veruska M. Costa, Alberto N. Ramos Jr, Andrea S. Souza, Yara M. Gomes.

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
