## [Decision Letter · Decision Letter 0]

10 Mar 2020

Dear PhD Santos,

Thank you very much for submitting your manuscript "Acute Chagas disease in Brazil from 2001 to 2017: A nationwide spatiotemporal analysis" for consideration at PLOS Neglected Tropical Diseases. As with all papers reviewed by the journal, your manuscript was reviewed by members of the editorial board and by several independent reviewers. In light of the reviews (below this email), we would like to invite the resubmission of a significantly-revised version that takes into account the reviewers' comments. 

We cannot make any decision about publication until we have seen the revised manuscript and your response to the reviewers' comments. Your revised manuscript is also likely to be sent to reviewers for further evaluation.

Sincerely,

Igor C. Almeida

Associate Editor

Alain Debrabant

Deputy Editor

Reviewer's Responses to Questions

**Key Review Criteria Required for Acceptance?**

**Methods**

-Are the objectives of the study clearly articulated with a clear testable hypothesis stated?

-Is the study design appropriate to address the stated objectives?

-Is the population clearly described and appropriate for the hypothesis being tested?

-Is the sample size sufficient to ensure adequate power to address the hypothesis being tested?

-Were correct statistical analysis used to support conclusions?

-Are there concerns about ethical or regulatory requirements being met?

Reviewer #1: The objectives are clear and in accordance with the retrospective study design. I think the sample size is sufficient is suitable to the tested hypothesis and statistical analysis supports the conclusions. Regarding ethical principles, there are no concerns.

Reviewer #2: Please see general comments below.

**Results**

-Does the analysis presented match the analysis plan?

-Are the results clearly and completely presented?

-Are the figures (Tables, Images) of sufficient quality for clarity?

Reviewer #1: The results correspond to the plan and the data are clear and complete. Figures and tables are self-explanatory and quality.

Reviewer #2: Please see general comments below.

**Conclusions**

-Are the conclusions supported by the data presented?

-Are the limitations of analysis clearly described?

-Do the authors discuss how these data can be helpful to advance our understanding of the topic under study?

-Is public health relevance addressed?

Reviewer #1: The conclusions are appropriate to the data emphasizing the study limitations due to the quality of the analyzed data and the underreporting of Trypanosoma cruzi in the acute phase. Data analysis is well discussed considering epidemiological indicators related to the acute phase of Chagas disease. It also addresses the importance of reporting suspected cases in primary care and public health services, early diagnosis as well as the gaps regarding the availability of the etiological treatment as relevant factors in controlling the transmission of the parasite.

Reviewer #2: Please see general comments below.

**Editorial and Data Presentation Modifications?**

Reviewer #1: I suggest reviewing the entire manuscript of the text to make the changes below:

Abstract: Lines 32, 35, 38 41 and 51. Change the acronym CD to ACD so as not to confuse the reader, CD is Chagas disease and ACD is acute Chagas disease.

Author summary: Line 63 - Put number 4 in full (numbers 0-10 must be written in full). 

Lines 69, 70, 73 and 79 - Change the acronym CD to ACD.

Introduction: Lines 106, 110 and 115 - Change the acronym CD to ACD.

Material and Methods: Lines 144, 146, 152, 162 and 176 - Change the acronym CD to ACD.

Results: Lines 195, 196, 203, 213, 217 and 243 - Change the acronym CD to ACD.

Fig 2 line 207 and Fig 4 line 268 - Do the same.

Review legend of Fig. 3 - No map self-reported skin color - Please review Fig 3 - in blue, white is misspelled, rewrite!

Please Review the list of references, space between words, names of authors etc.

See manuscript attached.

Reviewer #2: Please see general comments below.

**Summary and General Comments**

Reviewer #1: This study adresses epidemiological data related to acute Chagas disease that are publicly available, analyzed the spatiotemporal distribution of notified cases of acute Chagas disease and evaluated relevant epidemiological indicators throughout Brazil from 2001 to 2017. Even with a huge difference between the number of confirmed cases of acute Chagas disease and the notification (suspected) cases reported, there was a clear increase in numbers of acute Chagas disease cases during the last decade. The importance of diagnosis and prompt treatment of cases and prolonged follow-up of patients should be strengthened in the increased risk of outbreaks, providing visibility and development of methodologies and well-designed clinical follow-up.

Reviewer #2: This is a sound analysis of acute cases of Chagas Disease during a period of 17 years. Brazil's macro regions were defined as units of analysis. Methods and conclusions are adequate. There are only some comments:

I am missing a general description of all notified cases and request to include a table with all cases diagnosed, according to available variables available such as sex, age, region, transmission route, year/period etc. (Figure 3 only gives a broad information by region/bar charts).

Acute Chagas Disease often occurs in outbreaks, especially in the case of oral transmission. In addition, a major outbreak may distort geographical analysis. I suggest to include information on major outbreaks that occurred during the study period, in the discussion section of the manuscript, and to discuss the implications adequately.

This manuscript is written by 15 authors, which is rather unusual for a study analyzing easily available secondary data. However, a description of author contributions is missing. Please include. 

Minor comments:

Lines 85/86: Correct ... “by the Kinetoplastid hemoflagellated Trypanosoma cruzi.” Into “by the hemoflagellated kinetoplastid Trypanosoma cruzi”. 

Line 301: should be “previous study” – please correct

PLOS authors have the option to publish the peer review history of their article (what does this mean?). If published, this will include your full peer review and any attached files.

Reviewer #1: No

Reviewer #2: No
---

## [Decision Letter · Decision Letter 1]

17 Apr 2020

Dear PhD Santos,

Thank you very much for submitting your manuscript "Acute Chagas disease in Brazil from 2001 to 2017: A nationwide spatiotemporal analysis" for consideration at PLOS Neglected Tropical Diseases. As with all papers reviewed by the journal, your manuscript was reviewed by members of the editorial board and by several independent reviewers. The reviewers appreciated the attention to an important topic. Based on the reviews, we are likely to accept this manuscript for publication, providing that you modify the manuscript according to the review recommendations. 

Sincerely,

Igor C. Almeida

Associate Editor

Alain Debrabant

Deputy Editor

Reviewer's Responses to Questions

**Key Review Criteria Required for Acceptance?**

**Methods**

-Are the objectives of the study clearly articulated with a clear testable hypothesis stated?

-Is the study design appropriate to address the stated objectives?

-Is the population clearly described and appropriate for the hypothesis being tested?

-Is the sample size sufficient to ensure adequate power to address the hypothesis being tested?

-Were correct statistical analysis used to support conclusions?

-Are there concerns about ethical or regulatory requirements being met?

Reviewer #1: Accept

Reviewer #2: (No Response)

**Results**

-Does the analysis presented match the analysis plan?

-Are the results clearly and completely presented?

-Are the figures (Tables, Images) of sufficient quality for clarity?

Reviewer #1: (No Response)

Reviewer #2: Table S2: These are the numbers as used for the Figure. Please include a standard table in the manuscript presenting data of all cases, without stratification by region and period! Include relative frequencies! Variables: age, gender, skin color, route of infection, region, year.

**Conclusions**

-Are the conclusions supported by the data presented?

-Are the limitations of analysis clearly described?

-Do the authors discuss how these data can be helpful to advance our understanding of the topic under study?

-Is public health relevance addressed?

Reviewer #1: (No Response)

Reviewer #2: (No Response)

**Editorial and Data Presentation Modifications?**

Reviewer #1: Accept

Reviewer #2: (No Response)

**Summary and General Comments**

Reviewer #1: (No Response)

Reviewer #2: In general, the manuscript is acceptable. However, inclusion of the table with description of the cases, as described above, is crucial.

PLOS authors have the option to publish the peer review history of their article (what does this mean?). If published, this will include your full peer review and any attached files.

Reviewer #1: No

Reviewer #2: No
---

## [Editor Report · Decision Letter 2]

1 Jun 2020

Dear PhD Santos,

Thank you very much for submitting your manuscript "Acute Chagas disease in Brazil from 2001 to 2018: A nationwide spatiotemporal analysis" for consideration at PLOS Neglected Tropical Diseases. As with all papers reviewed by the journal, your manuscript was reviewed by members of the editorial board and by several independent reviewers. The reviewers appreciated the attention to an important topic. Based on the reviews, we are likely to accept this manuscript for publication, providing that you modify the manuscript according to the review recommendations. 

Associate Editor's comments:

I think the authors did a very good job in addressing the major concerns of the reviewers in this new R2 version. However, I have some concerns myself that need to be clarified before acceptance of this manuscript.

Figure 3: I noticed that the revised graph shows datapoints from 2001 to 2017. However, the legend (line 207) states the data are from 2001 to 2018. The same issue applies to Figure 4, where the new maps are from 2002 to 2017, whereas 2018 is missing. On the other hand, Table 1 has data from 2001 to 2018. I understand that some of the official parameters are perhaps not available for 2018 or any other particular year(s), but the manuscript needs to keep a certain consistency throughout it. Please, clarify these discrepancies in the text, just stating when and, if possible, why certain parameter(s) or data for a particular year is/are not available.

Sincerely,

Igor C. Almeida

Associate Editor

Alain Debrabant

Deputy Editor

Associate Editor's comments:

I think the authors did a very good job in addressing the major concerns of the reviewers in this new R2 version. However, I have some concerns myself that need to be clarified before acceptance of this manuscript.

Figure 3: I noticed that the revised graph shows datapoints from 2001 to 2017. However, the legend (line 207) states the data are from 2001 to 2018. The same issue applies to Figure 4, where the new maps are from 2002 to 2017, whereas 2018 is missing. On the other hand, Table 1 has data from 2001 to 2018. I understand that some of the official parameters are perhaps not available for 2018 or any other particular year(s), but the manuscript needs to keep a certain consistency throughout it. Please, clarify these discrepancies in the text, just stating when and, if possible, why certain parameter(s) or data for a particular year is/are not available.
---

## [Editor Report · Decision Letter 3]

2 Jun 2020

Dear PhD Santos,

We are pleased to inform you that your manuscript 'Acute Chagas disease in Brazil from 2001 to 2018: A nationwide spatiotemporal analysis' has been provisionally accepted for publication in PLOS Neglected Tropical Diseases.

Best regards,

Igor C. Almeida

Associate Editor

Alain Debrabant

Deputy Editor

---

## [Editor Report · Acceptance letter]

13 Jul 2020

Dear PhD Santos,

We are delighted to inform you that your manuscript, "Acute Chagas disease in Brazil from 2001 to 2018: A nationwide spatiotemporal analysis," has been formally accepted for publication in PLOS Neglected Tropical Diseases.

Best regards,

Shaden Kamhawi

co-Editor-in-Chief

Paul Brindley

co-Editor-in-Chief
